# 3D Sugar Printing of Networks Mimicking the Vasculature

**DOI:** 10.3390/mi11010043

**Published:** 2019-12-30

**Authors:** Andreas M. A. O. Pollet, Erik F. G. A. Homburg, Ruth Cardinaels, Jaap M. J. den Toonder

**Affiliations:** 1Microsystems, Department of Mechanical Engineering, Eindhoven University of Technology, 5600MB Eindhoven, The Netherlands; a.m.a.o.pollet@tue.nl (A.M.A.O.P.); f.g.a.homburg@tue.nl (E.F.G.A.H.); 2Institute for Complex Molecular Systems, Eindhoven University of Technology, 5600MB Eindhoven, The Netherlands; 3Polymer Technology, Department of Mechanical Engineering, Eindhoven University of Technology, 5600MB Eindhoven, The Netherlands; r.m.cardinaels@tue.nl

**Keywords:** 3D printing, sugar glass, microfluidics, microfabrication, microvasculature, flow analysis

## Abstract

The vasculature plays a central role as the highway of the body, through which nutrients and oxygen as well as biochemical factors and signals are distributed by blood flow. Therefore, understanding the flow and distribution of particles inside the vasculature is valuable both in healthy and disease-associated networks. By creating models that mimic the microvasculature fundamental knowledge can be obtained about these parameters. However, microfabrication of such models remains a challenging goal. In this paper we demonstrate a promising 3D sugar printing method that is capable of recapitulating the vascular network geometry with a vessel diameter range of 1 mm down to 150 µm. For this work a dedicated 3D printing setup was built that is capable of accurately printing the sugar glass material with control over fibre diameter and shape. By casting of printed sugar glass networks in PDMS and dissolving the sugar glass, perfusable networks with circular cross-sectional channels are obtained. Using particle image velocimetry, analysis of the flow behaviour was conducted showing a Poisseuille flow profile inside the network and validating the quality of the printing process.

## 1. Introduction

Creating faithful models of the vasculature is key for obtaining new and important insights in how the vasculature functions and plays its role as distribution network inside the body. Being able to implement the vasculature in “Organ-on-Chip” models is essential for improving the quality and functionality of these models [1]. Without a vasculature system adequate supply of oxygen and nutrients will be limited, keeping the scale of the model within the diffusion limit [2]. The architecture of the vasculature differs also greatly between different organs, and the ability to match the in vivo design can increase the biological relevance of the Organ-on-Chip model [3]. Similar, the vascular network also changes under influence of many diseases [4,5]. This change can be either in properties of the blood flow (e.g., due to changed permeability of the vessel wall), changes in the geometry (e.g., the development of tortuosity), or both [6,7]. Modelling these parameters can lead to a better understanding of disease progression and associated effects. Subsequent treatment of the disease is most commonly also via the vasculature, therefore understanding drug distribution and local concentration is important for effective treatment. Development of nano- and micro-particle drug delivery for local treatment is becoming more common, and combining this with a better understanding of particle-flow behaviour enables a more effective treatment of the disease [8,9].

Mimicking the microvasculature remains one of the challenging aspects in microfabrication of biomedical devices such as Organ-on-Chip [10]. This is mainly due to the circular cross-section of the channels, the wide range of diameters ranging from 1 mm down to 5 µm and the often complex network architecture [1]. Standard fabrication methods fail to represent all these features faithfully. A range of dedicated fabrication methods have already been developed, however a method that includes all capabilities and strengths is still missing. Laser ablation can be used to create round channels of all sizes present in the vasculature in a large range of hydrogels [11]. This method is reproducible and robust, capable of creating copies of networks even in Organ-on-Chip applications [12]. Fabrication of especially larger networks and diameters is however time consuming and an expensive setup is required. Photolithography methods that are most commonly used in microfabrication have the capability of creating channels across length scales and complex networks [13,14]. Standard photolithography techniques result however in 2D rectangular channel geometries rather than circular channels resulting in a different flow profile and particle distribution than normally found in vivo. Additional multi-layered techniques can create more rounded geometries and a larger variation in channel diameter [14]. This requires however alignment of the different layers, as well as repeating the process multiple times for creating a round vascular network. Viscous finger patterning can be used to change the rectangular geometry obtained with photolithography into a rounded geometry [15]. This technique can work for different diameters in one network but only bifurcations can be made, whereas merging channels are not possible with this technique. 3D printing is becoming a more common and standard technique, showing capabilities for fabricating complex structures quickly and with low effort. This technique enables to recreate complex networks, however standard 3D printing limits the choice of materials and quality of prints [16]. This leaves the process restricted to specific materials that are mostly not biocompatible. Changing to biocompatible materials makes this compatible with Organ-on-Chip applications but leaves the material choice and processing method restricted [17,18]. 

3D printing of sugar glass however holds great promise to recreate the microvasculature structures [19,20,21]. Others have shown that it is possible to fabricate networks that can be cast in a variety of materials and that are perfusable after dissolving the sugar glass [19,20]. These methods individually show the capabilities for fabricating a certain aspect of the vasculature, however a method enabling control over the combination of size, geometry and complexity is still missing. The work by Miller et al. showed the first promising application of sugar printing for creating network like structures [19]. These however were limited to lattice networks with fibres on top of each other to form junctions. This is a disadvantage since junction geometry is essential for a physiological representation of vascular networks. Work by Gelber et al. improved the printing process considerably and performed an in depth analysis of the printing behaviour, showing high resolutions and small diameters [21]. Printed structures however maintained lattice based and the application as a vascular network was still missing.

Therefore, our main goal was to engineer hierarchical 3-dimensional branching networks of channels having circular cross section, with full control over diameter. This means that we should be capable of: fabricating both in-plane bifurcations and merging points;implementing this multiple times to form complex networks;having control over fibre diameter not only per fibre but also during printing;printing both geometric shapes as well as organic shapes.

In this paper we demonstrate the ability of creating bifurcating networks with in-plane bifurcations as well as control over fibre diameter during printing using a custom-made 3D pressure extrusion printer. Additionally, the flow within the networks was analysed using particle image velocimetry, demonstrating the capability of investigating the flow behaviour and validating the quality of the printed networks. To the best of our knowledge, this is the first study reporting the fabrication of 3D sugar printed vascular flow models that combine all the properties mentioned, and that are characterized using a quantitative flow analysis.

## 2. Materials and Methods 

### 2.1. Printing Setup

We built a dedicated 3D printing setup that is based on the principle of pressure extrusion as seen in Figure 1. 

The base is an optical table (TMC, Peadbody, MA, USA with a vertical linear stage (110150, Newport, Irvine, CA, USA) as the Z-axis. The X-axis is mounted on the translating Z-axis and the Y-axis to the base plate, both consisting of aluminium construction profiles with belt driven linear sliders. The extrusion is implemented using compressed air with a switched pressure controller (EFD performus III, Nordson, East Providence, RI, USA). A switching valve (FHT Perslucht, Deurne, The Netherlands) is used to select the active extruder. Two extruders are used with a different nozzle diameter to increase the printable range and decrease printing time. Heating mantles are made out of aluminium with two 40 W heater cartridges each. These were designed specific for this case with a high thermal mass and uniform temperature distribution, resulting in a stable temperature during extrusion. Tightly fitting barrels that can be equipped with 3D printing nozzles (E3D, Chalgrove, UK) are vertically kept in place via a V-groove and set screw. For cooling the extruded fibres a fan is mounted at one end of the Y-axis at a fixed distance from the nozzle. Software used is Marlin with Repetier-Host [22,23], using the auxiliary ports on the board for activating and switching the extrusion process. Glass slides (50 × 75 mm, Corning, Corning, NY, USA) are used as substrate for printing, which are kept in place using a vacuum chuck. All components, unless stated otherwise, are off-the-shelf 3D printer components (123-3D.nl, Almere, The Netherlands). A full list of materials as well as drawings, detailed pictures and software code, are available in the Appendix A.

### 2.2. Carbohydrate Glass Preparation and Characterisation

We used two types of blends in our experiments, sugar glass and carbohydrate glass. Sugar glass was prepared by dissolving 53 g of sucrose (S9378, Sigma-Aldrich, Darmstadt, Germany) and 25 g of glucose (G8270, Sigma-Aldrich) in 50 mL water (MilliQ, Darmstadt, Germany) and heating it to 150 °C on a stirring hot plate with external thermometer (ETS-D5, IKA RH, Staufen, Germany) to remove all water [19,20,21]. The carbohydrate glass was made by adding, to the sugar glass, an additional 10 g of dextran with an average molecular weight of 48–90 kg/mol (D3759, Sigma-Aldrich) and 50 mL water (MilliQ) to fully dissolve the mixture before evaporating the water [19]. Once the solution stopped boiling it was poured in preheated barrels (120 °C) with nozzles already fitted, to prevent the formation of air bubbles. The prepared blends are hygroscopic and were therefore stored in a closed box with silica beads (VWR, Amsterdam, The Netherlands).

Characterisation of the carbohydrate blends was done using a rheometer (RDA III, Rheometrics, Frankfurt, Germany) with a 50 mm flat bottom plate and 25 mm flat top plate. Prepared samples (25 mm diameter × 2 mm height) were loaded at a temperature of 80 °C before measuring. A nitrogen-flushed convection oven was used to avoid absorption of moisture from the air during measurements. Oscillatory measurements (1 Hz) in the linear viscoelastic regime (1% strain for sugar glass and 0.1% for carbohydrate glass) were performed to determine the storage and loss moduli as function of temperature. Continuous shear experiments were performed to determine the temperature and shear rate dependency of the viscosity. A temperature ramp rate of 1 °C/min was used and for the shear rate dependency a 10 s delay was applied before 30 s of measuring at each shear rate. 

Fibre diameters were investigated by measuring freely suspended fibres printed over a frame, at 7 points per fibre 2 mm apart and 1 mm from the frame using an optical measuring microscope (NEXIV VMZ-R3020, Nikon, Leuven, Belgium) with automatic edge detection. These measurements were performed for fibres printed with different translation speeds, temperatures and blends to characterize the dependency on printing parameters. Data are plotted as mean with standard deviation for 5 fibres on the same frame printed in both directions. 

### 2.3. Fibre and Network Fabrication

For printing the sugar glass a pressure of 1 bar and a temperature of 90 °C were used unless stated otherwise. As a starting point a supporting rectangular framework of sugar glass (20 × 15 × 2 mm, thickness 1 mm) was printed on a glass slide (50 × 75 mm, Corning). Freely suspended fibres are printed across this framework, either resulting in single fibres, or forming networks. The latter are created by printing multiple intersecting single fibres, and junctions are created by reflowing, i.e., by locally heating the corresponding intersection with the heated nozzle, followed by cooling. Merging of fibres was done in a similar way by reflowing and subsequently moving the nozzle towards the end of the fibre network without extrusion. Air cooling was used to solidify the fibres during printing and to retain geometry when reflowing junctions. G-code of printed structures, including explanation, is available in the Appendix A. Printed structures needed to be stored under low humidity environment to preserve shape and structure.

Printed structures were cast in polydimethylsiloxane (PDMS 1:10, Sylgard 184, Dow Corning, Midland, MI, USA) and cured under vacuum for 48 h. The carbohydrate prints were dissolved from the PDMS cast by submerging the PDMS cast in water for 24 h, leaving a perfusable network.

Networks of square channels were made by creating a casting mould out of polymethylmetacrylate (PMMA, Eriks, Alkmaar, The Netherlands) using a CNC milling machine (MXD-40A, Roland, Irvine, CA, USA) with a 0.5 mm end mill (HPTEC 818, HPtec GmbH, Nieuwe Dorp, The Netherlands). The overall design was based on one of the sugar printed networks, but having a 500 × 500 µm square channel cross section rather than a circular cross section (see Appendix A). This casting mould was filled with PDMS and cured at 65 °C for 2 h; subsequent bonding of the PDMS channel structures to glass slides (25 × 75 mm, VWR) resulted in the formation of a perfusable network.

### 2.4. Flow Experiments

The flow field within the network was characterised with Particle Image Velocimetry (PIV) using an Axio Observer A1 (Zeiss, Oberkochen, Germany) with a 5× 0.16 NA objective (EC Plan-Neofluar, 420330-9901, Zeiss) and a Nano L 50–100 laser (532 nm, Litron Optical, Warwickshire, UK). 2 µm red fluorescence polymer microsphere particles (R0200, Duke Scientific, Fremont, CA, USA) at a concentration of 2.5 × 10^−3^ per mL were used as tracer particles, dispersed in water. The water was pumped through the network with a syringe pump (Fusion 200, Chemyx, Stafford, TX, USA) at a flow rate of 100 µL/min. 300 image pairs (900 µs interval) were taken at 10 Hz with an Imager pro camera (LaVision, Bicester, UK). Using Davis 7.2 (LaVision) the velocity vector field was determined within the network channels using the sum of correlation function, with a final resolution of 4 × 4 pixels (7.2 × 7.2 µm). To exclude background noise and resulting stray vectors, the masking function was applied in order to only include the channel area. 

## 3. Results

### 3.1. Rheology Results

Rheology measurements were performed for both blends to better understand their properties and the corresponding printing behaviour at different temperatures, as well as to investigate the role of adding dextran. The results are shown in Figure 2. 

Figure 2a depicts the complex viscosity (η*) as a function of temperature for both blends at 1 Hz. Clearly the viscosity decreases with temperature, and the carbohydrate glass viscosity is larger than that of the sugar glass. The latter can be explained, when considering the glass transition temperature (Tg) of the different components of the blends as found in literature [24,25,26], shown in Table 1. This shows that dextran has a higher Tg compared to sucrose and dextrose. Moreover, sucrose and dextrose are low molecular weight components (mono- and disaccharides) whereas dextran is a branched polysaccharide. It is therefore indeed expected that the carbohydrate glass, due to adding dextran, has an increased viscosity compared to sugar glass. Due to the higher viscosity of the carbohydrate glass compared to the sugar glass, we expect that a higher printing temperature or pressure must be used for the former than for the latter to obtain the same printing behaviour.

Figure 2b shows that, similar to the viscosity, both the storage (G’) and loss (G”) modulus are increased upon adding dextran to the blend (to obtain the carbohydrate glass) whereby G’ is even increased several orders of magnitude. This reflects the more elastic behaviour of the carbohydrate glass as compared to the sugar glass when printing at the same temperature. In general increasing the elasticity of the material leads to a higher melt strength, which means that a molten thread of material can be drawn with a higher force before breaking [27]. For printing small diameter fibres this is beneficial since the fibres are less prone to breaking or deformation. 

Besides giving information on the material behaviour during printing we can also find properties for subsequent storage and processing of the sugar prints from the rheology data. Important to state here is that the behaviour of the sugar glass is greatly influenced by humidity [25]. Storing the prints in normal air for a couple of minutes can already lead to a shift in Tg, resulting eventually in sagging of the structures or breakup of the fibres (Appendix B). Additional experimental results of viscosity and shear rate dependency of the different sugar and carbohydrate glass blends can be found in Appendix C. These results show that the viscosity of sugar glass is constant and that carbohydrate glass has a shear thinning behaviour.

### 3.2. Printing Behaviour

To have control over the final fibre diameter, it is essential to know the effect of printing parameters on the printing behaviour. Therefore, we performed several measurements at different printing speeds and temperatures for both the sugar and the carbohydrate glass. Figure 3a shows an image of two printed carbohydrate glass fibres suspended on top of the pre-printed framework, showing a constant diameter across the full fibre. Figure 3b zooms in on the connection of a carbohydrate glass fibre with the frame, showing that some necking can occur at this location. Note, that these figures make clear how we measured the fibre diameter and the effect of necking at the start of the fibre, but they were not used for perfusion experiments or casting reported in Section 3.3 and Section 3.4. Figure 3c shows the fibre diameter for the carbohydrate glass as a function of nozzle translation speed, at different temperatures. The obtained diameter values range from 120 ± 12 µm to 569 ± 36 µm. The diameter decreases for increasing speed, and it increases with temperature. Figure 3d shows the fibre diameter for the sugar glass as a function of nozzle translation speed, at *T* = 90 °C. From a comparison between Figure 3c,d, it is clear that the diameter of the sugar glass fibres is substantially larger than that of the carbohydrate glass, at equal printing conditions. This confirms our expectation (see Section 3.1) that a higher printing temperature must be used for the carbohydrate glass than for the sugar glass to obtain the same printing behaviour. A more quantitative analysis can be done on the basis of theory from the literature on glass fibre drawing [19,20]. Based on mass conservation we expect that the fibre diameter (*D*_f_) follows: (1)$$D{\;}_{f}=\;\frac{2}{\sqrt{\pi }}\frac{\sqrt{Q}}{\sqrt{u}}\;$$
where *u* is the translation speed, and *Q* the flowrate through the nozzle. The latter is given by:(2)$$Q=\frac{\pi D_{n}{}^{4}}{128\;\eta \left(T\right)L_{n}}\Delta p\;$$
which includes: pressure applied over the nozzle (Δ*p*); nozzle diameter (*D_n_*); nozzle length (*L_n_*) and the temperature dependent viscosity (*η*(*T*)). Indeed, the dependency on *u* given by Equation (1) is confirmed by the results of Figure 3, by fitting the equation to the experimental data with $\sqrt{Q\;}$ as fitting parameter. Values for the fit are given in Table 2 along with the estimated values based on Equation (2). The quantitative agreement between the two is not perfect, but they show the same trend. Flow rate will increase when the viscosity becomes smaller (see Equation (2)), which happens when temperature increases as confirmed by Table 2. As a consequence, *Q* in Equation (1) increases and this predicts that at higher temperatures, larger diameter fibres are printed. This is also confirmed in Figure 3a. Finally, as we have seen in Figure 2a, the carbohydrate glass has a higher viscosity due to the addition of dextran; this higher viscosity results in a lower flow rate (see Equation (2)) and therefore, according to Equation (1), it should have a smaller fibre diameter than the sugar glass, which is also confirmed by Figure 3. 

What is also clear from Figure 3c is that an increase of temperature results in a higher standard deviation in fibre diameter. This is also found for fibres printed at a low speed of 25 mm/min. Using sugar glass instead of carbohydrate glass at the same temperature results in a loss of control over the fibre diameter and it becomes impossible to consistently print small diameter fibres, as can be seen from the large standard deviation in Figure 3d. This is most likely due to the fact that the extruded material builds-up on the nozzle, resulting in different apparent diameter and thus changes in fibre diameter. Changing to a larger nozzle diameter for printing larger diameter fibres is recommended to circumvent this problem and maintain accuracy. For the sugar glass it is better to print at a lower temperature, hence higher viscosity, to obtain a similar curve as seen with the carbohydrate glass in Figure 3c; the higher viscosity will likely not only lead to smaller diameters, but also lower variation. The change in blend to carbohydrate glass will also result in fibres that have a more consistent diameter over the full length of the fibre, less necking, because of the higher G’, as can be seen in Figure 3a,b.

### 3.3. Printing and Casting Network Structures

From the results obtained with respect to fibre printing behaviour, we opted for using the sugar glass at a temperature of 83 °C with a 0.15 mm nozzle at a translation speed of 50 mm/min for printing the network structures. Despite the better fibre formation with carbohydrate glass, sugar glass was used because of its clean start and stop behaviour during printing. The higher G’ of the carbohydrate actually hampers the formation of networks because discontinuation of the fibre is harder to achieve. This results in residual fibres being more common during printing. Furthermore, sugar glass showed better reflow capabilities compared to the carbohydrate glass because of the lower viscosity at printing temperatures, making it possible to create better defined in-plane junctions. The networks were all suspending on a pre-printed frame of sugar glass with a dimension of 20 × 15 mm and a height of 2 mm. During printing of these networks cooling is crucial to lower the temperature of the printed sugar glass below its Tg. Insufficient cooling will lead to sagging of the structures and loss of print geometry. By reflowing the sugar glass as explained in Section 2.3, in-plane junctions can be created were otherwise fibres would be stacked on top of each other and only partially melt together [19]. 

Figure 4a–d show top-view photos of printed networks (Figure 4a,c in which also the frame can be seen), as well as PDMS casts of the same networks (Figure 4b,d). Note, that the networks are suspended at a height of 2 mm above the glass substrate. The network shown in Figure 4a illustrates the possibility to connect adjacent fibres and form crossing channels, while Figure 4c shows a second order bifurcating network similar to a branching vasculature tree [28]. Casting in PDMS and subsequently dissolving the sugar glass leaves a circular open channel network as seen in Figure 4b,d. Figure 4e–h show perspective and detail images of printed networks. Figure 4e is a detailed image of the bifurcation, showing the in-plane bifurcation that has completely flowed together to form a monolithic structure. Figure 4f is an example of a more complex freeform 3D structure that can be printed. The structure consists of two fibres with bifurcations that are in the same plane, one diagonal fibre curving upwards and one straight fibre curving downwards. This example shows that it is possible to fabricate curved structures in 3D. Figure 4g is a curved trifurcation with all fibres having an equal curvature; the two outer fibres are in the same plane while the middle fibre curves upward. Figure 4h is a detailed image of the trifurcation point, again showing how the fibres flow together to form a monolithic structure with a smooth surface. Additional network prints and casts can be found in Appendix D.

### 3.4. Flow Experiments

Using the created networks, we analysed the flow behaviour using PIV. Results are shown in Figure 5 for different locations for both the networks of Figure 4b,d. All results approximately have the parabolic Poisseuille flow profile that we expected because of the circular cross-sectional channel geometry and the laminar flow inside the channels. The laminar behaviour of the flow is especially clear in Figure 5a where the channels cross. For the bifurcation in Figure 5b, a drop in absolute velocity can be seen after the bifurcation because of the splitting of the flow in two similarly sized channels to the channel before the bifurcation. This same effect is found in the second order bifurcating network as seen in Figure 5c,d. The absolute velocity further drops at the second order bifurcation seen in Figure 5c, when compared to the first order bifurcation in the same network as seen in Figure 5d. Figure 6a depicts the flow profile along the cross-section at the inlet from Figure 5b, averaged between Y = 0.18 and 0.37 mm along with the corresponding standard deviation. The averaged profile reasonably matches the profile calculated by using the following Equation (3), for the Poisseuille profile of a circular channel of the same diameter:(3)$$V\left(r\right)=\frac{2Q}{\pi r_{c}^{4}}\left(r_{c}^{2}-r^{2}\right),$$
where *r_c_* is the channel radius, and *r* the radial coordinate. These results all validate the use of these network models for in-depth investigation of the flow and also give a quantitative analysis of the printing quality, especially at the bifurcations where the flow shows a smooth transition (for example in Figure 5b). The full PIV analysis of both the crossing and second order bifurcating network can be found in Appendix E.

Additional analysis of the flow and a comparison between a circular and square cross-sectional channel is shown in Figure 6b. What can be observed is that, in the midplane, the velocity profile in the square channel is parabolic and agrees with the profile of a circular channel that has the same cross sectional area at equal flow rate. However, the velocity profile increasingly deviates from a parabolic profile when we move away from the midplane, which is distinctly different for a circular channel where the parabolic profile is maintained. This is also reflected in the distribution of the velocities over the whole channel cross section, where the lower velocities are more prominent in a square channel (see Figure A6, Appendix F). The difference in velocity profiles will substantially affect the distribution of particles within the channels [29]. Experimental data for the flow at the bifurcation within square channels can be found in Appendix G, showing an increased area of low velocities at the bifurcation point when moving out of the midplane, which is a cause of particle stagnation at such locations.

To demonstrate the importance of the channel geometry, we also present a numerical comparison between a circular and square cross-sectional channel of similar dimensions (diameter 564 µm vs. 500 × 500 µm) and with the same flow rate (100 µL/min); see Appendix F for the full analysis. The most striking difference is the shear stress profile on the walls of the channel as can be seen in Figure 7. In a circular cross-sectional channel, the wall shear stress is constant and independent of position on the wall, while a square channel shows a flattened parabolic profile with vanishing shear stress in the corners of the channel. This difference can have a number of significant effects that make the square channel less suitable for a vascular model: margination of cells/particles inside the flow can occur; the shear stress profile has effects on particle-wall binding/capturing; when culturing endothelial cells in the network, a non-physiological shear stress profile will affect cell behaviour and activation, as well as disease progression [29,30,31]. Moreover, the velocity field shows a different profile and distribution within square channels which affects transport of particles and molecules, for example resulting in particle stagnation in the corners of the square channel, which has been also found experimentally [32]. These effects result in a completely different, non-physiological distribution and flow of particles throughout the network, underlining the importance of creating circular cross sections for experimental vascular models [33].

## 4. Discussion and Conclusions

We have demonstrated the creation of simple microfluidic models as a step towards realistic models of the microvasculature, using in-air 3D sugar printing with a custom-built printer. The realised networks have fully controlled geometry consisting of circular cross-sectional channels with a diameter between 120 ± 12 µm and 569 ± 36 µm, and can contain bifurcating and merging channels. Previous work showed either control over geometry with large diameter fibres or small diameter fibres forming rudimentary networks [19,20,21]. In the work by Miller et al. [19] the printed structures were only lattice like and showed out of plane junctions, whereas in our work the control over the printing process was improved to obtain more freedom in design and enable the realisation of smooth in-plane junctions. The work by Gelber et al. [21] investigated the printing process extensively and shows high levels of control over the printing process by using more advanced controls and prediction algorithms compared to our setup. The prints that were made by Gelber et al. [21] are all lattice based structures of smaller diameter fibres that are based on a node to node printing procedure. In our work the printing is performed such that also curved and freeform organic shapes are possible. In our approach, bifurcations are formed by reflowing and cooling existing fibres in a controlled manner, creating a fused monolithic structure with no additional material and a smooth transition as validated by the PIV data. Our method shows that it is possible to create networks with small diameter channels and have freedom in design. By further improving the methods reported here it will be possible to print fibres down to ~100 µm and create more complex networks. The final goal would be to copy existing networks found in vivo, and use these for analysis and validation of existing models [34,35]. This application of freeform 3D sugar printing for fabricating vascular flow models makes our current approach unique and shows it is capable to form complex structures mimicking the vasculature. 

Rheological measurements provided insights into the material behaviour relevant for printing. These measurements showed that changing the blend composition can indeed influence viscosity and viscoelastic behaviour of the fibres during printing and cooling. Changes in blend composition will also lead to a different Tg that determines the maximum processing temperature for casting and the hygroscopic properties of the printed structures. Further research should look into a method for determining the Tg of the blend as well as a method for determining the water content of the sugar glass blends, verifying the expected little to no water content. With these methods at hand a more extensive investigation can be performed, which could include a wider variety in molecular weight of dextran as well as sugar alcohols and their effect on the blend and associated printing properties [21,24,25,26].

Fibre quality and stringing properties are dependent on the rheological properties of the blend and printing parameters, including temperature and nozzle translation speed. The ideal blend would have the following properties: high Tg, robust small diameter fibre formation and a clean start/stop of printing. As seen in the data it is difficult to achieve all properties in a single blend: carbohydrate glass shows robust fibre formation at small diameters, but poor control over start/stop properties, whereas sugar glass exhibits the clean start/stop properties, but less control over fibre diameter. Therefore a combinatory approach could be used to benefit from the strengths of both blends. By using the sugar glass to form the backbone network and geometry as shown in this paper, a large diameter basic network can be printed. This network can later be extended with small diameter fibres using the carbohydrate glass to create a multiscale network. By changing the translation speed during printing, the fibre diameter can be tuned, which is needed for implementing scaling laws for channel diameters as found in the in vivo vasculature [28]. Implementing these adaptations will enable more complex networks with more design freedom, creating the ability to form networks of specific cases including stenosis and aneurysms. 

Casting networks was now done in PDMS but we also were able to cast the networks in different materials such as agarose, gelatine, alginate and polyacrylamide similar to the work by Miller et al. [19] (data not shown). This will extend the possible applications towards different biological fields such as angiogenesis, vessel permeability and interstitial flow [36,37,38]. Increasing the biological relevancy of the model is crucial for creating models that are better in mimicking the in vivo microvasculature and for the ability of translating these results to in vivo cases [39].

Analysing the networks using PIV gave detailed information on the flow behaviour. Using blood, rather than water, as the flow medium will result in a flow behaviour similar to what is found in vivo. This is caused by increased viscosity of the medium and the dependency of this viscosity on vessel diameter, also known as the Fåhraeus-Lindqvist effect [40,41,42]. This can lead to a better insight in how particles are distributed across the network, for instance when applying a bolus injection, or in particle-blood interactions [9,39]. When combined with endothelial cells lining the inside of the channels, adhesion of particles and binding efficiency of targeted particles can be investigated [43]. For such studies, it is essential that the channel cross sections of the experimental vascular model are circular rather than rectangular.

Overall, 3D sugar printing shows to be a viable tool for creating networks with circular cross-sectional channels that mimic the microvasculature. The freedom in design and the biocompatibility of the method make it a versatile platform that can be adapted in a straightforward manner to different fields depending on the application in mind [30,44]. Next, we will use the technique to create realistic models of the vasculature for investigating flow and particle behaviour. 

## Figures and Tables

**Figure 1 micromachines-11-00043-f001:**
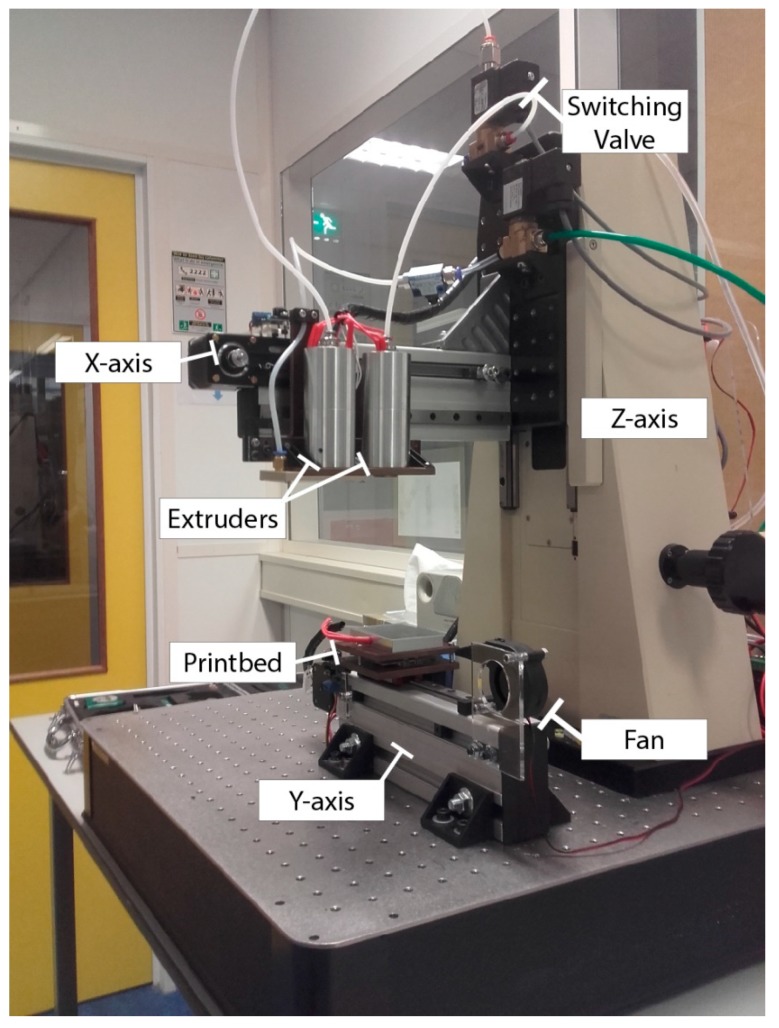
Dedicated 3D printer setup with pressure extrusion. Optical table has the Z and Y-axis mounted onto it, and the X-axis is mounted on the translation platform of the Z-axis. Y-axis consists of a belt driven slider with a vacuum chuck that can be levelled by three set screws and springs. X-axis has two extruders with two 40 W heater cartridges each. Extrusion can be switched using the valve mounted on the Z-axis and is controlled by a pressure controller (not shown in this figure). A fan is added to the end of the Y-axis to provide cooling of the extruded fibre during printing.

**Figure 2 micromachines-11-00043-f002:**
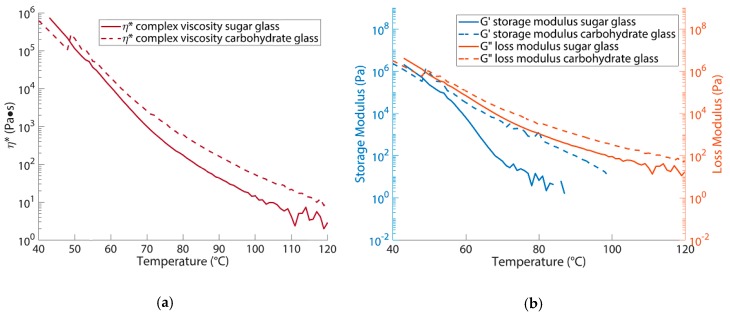
Rheological behaviour of sugar and carbohydrate glass: (**a**) Complex viscosity (η*) on a logarithmic scale for both blends as function of temperature measured at a frequency of 1 Hz. The carbohydrate glass shows a higher viscosity compared to the sugar glass—important for the temperature during fibre printing; (**b**) Storage (G’) and Loss (G”) modulus on a logarithmic scale for both blends as function of temperature. The storage modulus (G’) of carbohydrate glass is increased several orders of magnitude, making fibres less prone to breaking and necking during printing.

**Figure 3 micromachines-11-00043-f003:**
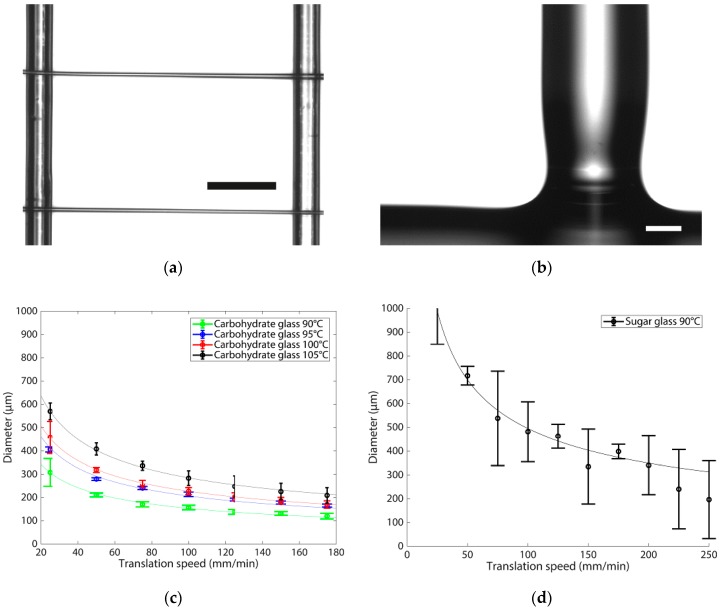
Relation between printed fibre diameter, nozzle translation speed and temperature for the carbohydrate glass and sugar glass fibres printed with a 0.15 mm nozzle. (**a**) setup to determine fibre diameter with two printed carbohydrate fibres suspended on top of the pre-printed framework, scale bar 1000 µm; (**b**) Detailed image of the carbohydrate fibre connection with the framework, showing some degree of necking at the beginning of the fibre, scale bar 100 µm; (**c**) Fibre diameter for the carbohydrate glass at different temperatures and translation speeds; (**d**) Fibre diameter for the sugar glass at 90 °C, showing the effect of lower viscosity on printing behaviour and quality. The fitted function is Equation (1) for glass fibre drawing with *Q* as the fitting parameter, with values given in Table 2.

**Figure 4 micromachines-11-00043-f004:**
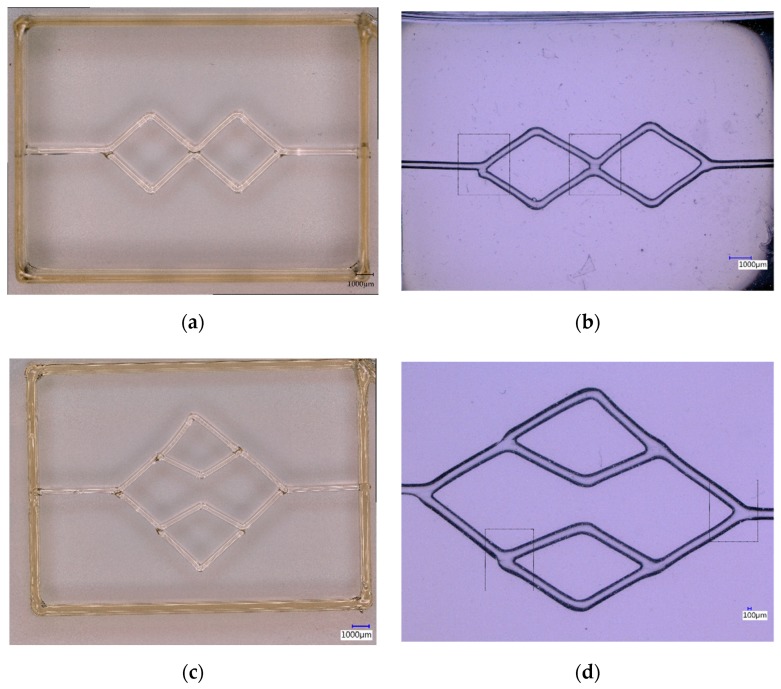
Suspended printed sugar networks and PDMS casts after dissolving the sugar glass: (**a**) Suspended crossing network of sugar glass printed on the framework; (**b**) PDMS cast after dissolving the sugar glass of a similar crossing network; (**c**) Second order bifurcating sugar glass network; (**d**) PDMS cast after dissolving the sugar glass of a similar second order bifurcating network. Dashed areas are for the PIV measurements in Figure 5; (**e**) Detailed image of the bifurcation, showing the monolithic in-plane structure of the circular cross-sectional fibres; (**f**) 3D sugar glass network structure with the straight fibre curving downwards and the diagonal fibre curving upwards; (**g**) Curved trifurcation sugar glass network with the middle fibre curving upwards, while other fibres are in-plane; (**h**) Detailed image of the trifurcation, showing the fusion of the three fibres into a single monolithic trifurcation. Scale bars for the perspective images are only valid for the focussed part.

**Figure 5 micromachines-11-00043-f005:**
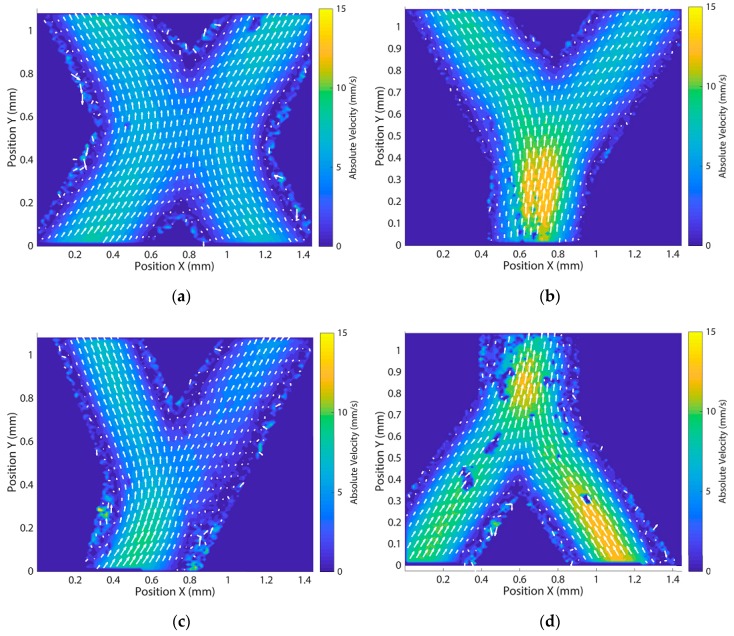
PIV measurement results for different crossings and bifurcations of the channel networks. Flow is from bottom to top of the images. Vectors indicate flow direction and absolute velocity. Absolute velocity is also represented in the colour coding of the flow field. (**a**) Flow direction and absolute velocity for the crossing junction inside the dashed area of the crossing network of Figure 4b; (**b**) Flow direction and absolute velocity for the bifurcating junction of the crossing network at the inlet (dashed area Figure 4b); (**c**) Flow direction and absolute velocity for the 2nd bifurcating junction of the second order bifurcating network in the dashed area of Figure 4d; (**d**) Flow direction and absolute velocity for the final merging junction of the second order bifurcating network inside the dashed area in Figure 4d.

**Figure 6 micromachines-11-00043-f006:**
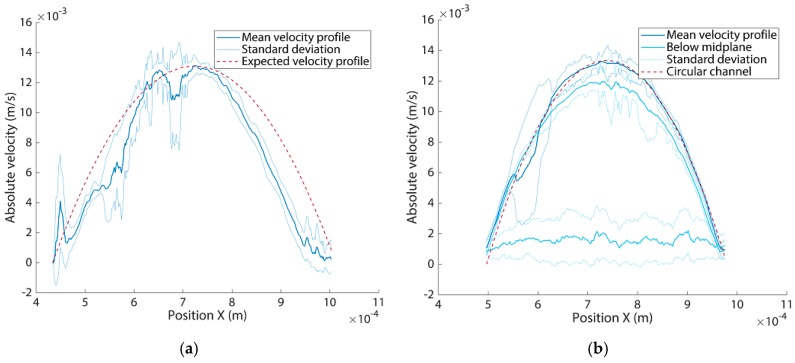
Averaged absolute velocity profile of the inlet channel of a bifurcating network for a circular and square cross-sectional channel in the mid-plane as well as below the midplane, at quarter height and near the bottom wall, for the square channel. (**a**) Averaged absolute velocity of the inlet channel of Figure 5b with standard deviation and the expected velocity profile based on Equation (3). Parameters used for Equation (3) are: *Q* = 100 µL/min and a channel radius, $\;r_{c}$, of 284 µm. Area under the curve for the expected velocity profile is 5.05 × 10^−6^ m^2^/s and for the measured velocity profile 4.77 × 10^−6^ m^2^/s; (**b**) Averaged absolute velocity profile with standard deviation of the square cross-sectional channel (500 × 500 µm) both in the mid plane and below the midplane, at quarter height and near the bottom wall in the channel. The expected velocity profile for a circular channel with equal cross-sectional area is based on Equation (3). Parameters used for Equation (3) are: *Q* = 100 µL/min and a channel radius, $\;r_{c}$, of 282 µm.

**Figure 7 micromachines-11-00043-f007:**
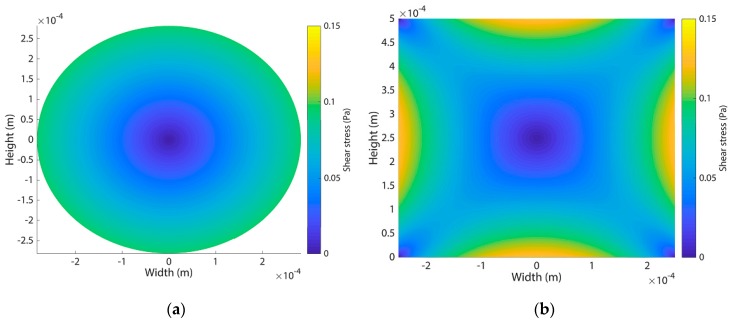
Theoretical shear stress profile in a circular and square cross-sectional channel of similar dimensions (diameter 564 µm vs. 500 × 500 µm) with a flow rate of 100 µL/min (**a**) Shear stress profile for a circular cross-sectional channel showing uniform shear stress along the wall and a radial symmetry; (**b**) Shear stress profile for a square cross-sectional channel showing a flattened parabolic profile along the wall with vanishing shear stress in the corners of the channel.

**Table 1 micromachines-11-00043-t001:** Glass transition (Tg) temperature for the different components present in the blends.

Component	Reported Tg	Reference
Sucrose	62–70 °C	[24,25,26]
Dextrose	31 °C	[25]
Dextran	150–225 °C *	[24,25,26]

* Depending on chain length of the dextran used.

**Table 2 micromachines-11-00043-t002:** Fitted (with Equation (1)) and estimated values (with Equation (2)) for $\sqrt{Q}$ for the carbohydrate glass at different temperatures, as well as the sugar glass.

Material	Temperature	$\mathrm{Fitted}\;\sqrt{Q}\;$	95% Confidence Interval	$\mathrm{Estimated}\;*\;\sqrt{Q}\;$
	°C	× 10^−6^	× 10^−6^	× 10^−6^
Carbohydrate glass	90	5.54	(5.45–5.63)	6.49
	95	7.46	(7.42–50)	8.60
	100	8.20	(8.09–8.31)	11.28
	105	10.29	(10.17–10.42)	14.4
Sugar glass	90	17.89	(17.34–18.43)	12.50

* With *D*_n_ = 0.15 mm, *L*_n_ = 0.18 mm, Δ*p* = 1 bar and η (T) from the complex viscosity at the given temperature, respectively: 163.9; 93.23; 54.26; 33.29 Pa·s for carbohydrate glass and 44.2 Pa·s for sugar glass.

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
