# Peer review of "3D Sugar Printing of Networks Mimicking the Vasculature"

_micromachines, 2019, doi:10.3390/mi11010043_

Round 1

Reviewer 1 Report

Reviewer’s Opinion

Although the authors' enthusiasm can be seen in this presented manuscript, it is hard to give a positive response as a current form due to the several problems. I think it would be good to correct some of the problems in your manuscript through a major revision. However, I'm worried that the points made in originality can be modified.

1) Originality: If the originality is to create the vascular network geometry using the sugar printing method, an explanation should be added to the difference from the existing photolithography. How do the results in Figures 5 and 6 differ from the results using conventional photolithograph microchannels? Frankly speaking, the results in Figure 5 and 6 are expected. In this case, the result of comparing the PDMS chip using silicon wafer with the sugar printing method suggested by the authors should be added.

2) Materials and methods: The company name and country of equipment and material should be addressed. For example, what is meaning of Line 97 (123-3D. nl) ?

3) Figure 2: I would recommend you to increase the sharpness of the figure image. If this is important, you had better give some additional explanation on why it is important.

4) Figure 3: Wouldn’t PDMS chip using silicon wafers produce better results?

5) Sentence correction: Please correct all sentences including "We". Everyone who reads already knows that this is the author's paper.

Reviewer 2 Report

The authors showed the ability to print 3D models to mimic microvasculature using sugar glass and carbohydrate glass. Authors built a custom 3D printer to perform this task. However, this paper lacks novelity and advances. Therefore, I recommend not to be considered for publication at this time. Printing 3D microvasculature structures using carbohydrate and sugar glass was showed before. More advanced 3D models with cells were shown in Nat Mater. 2012 Sep; 11(9): 768–774. New 3D printer for printing microvaculature using carbohydrate glass was demonstrated by Rohit Bhargava lab. This paper needs significant improvement with newer 3D models or biological applications before publication. 

Round 2

Reviewer 1 Report

I’ve read about your response and have some questions.

1) Originality: I still have questions about the uniqueness of this manuscript.

recommendation 1)

If the originality of this manuscript is different from the references (19-21) you mentioned in the main goal, I am oaky with your manuscript. Could you please remodified Line 399~Line 416 because it is very important parts in your manuscript ? Please add lines 83-92 to the originality of this paper using the following sentence.

To the best of our knowledge, it is the first study ~~~~ 

recommendation 2)

If the flow and distributions of particles were analyzed by an experimental model, the difference from the rectangular microchannel should be clarified by experiment. If the experiment is difficult due to several reasons, the references you mentioned should be compared and added their results into Figure 6 so that the originality of this manuscript can be differentiated from other papers. I think it will be difficult parts through a major revision, but I recommend you to follow my suggestion.

2) Please unify the round channels into circular channels in your manuscript (Line 58).

Reviewer 2 Report

Authors made significant improvements in the manuscript. It can be publishable in current form.

Author Response

We thank the reviewer for the positive recommendation!

Round 3

Reviewer 1 Report

I think you did best to follow my suggestion. I published mostly papers analyzing fluids in PDMS based on silicon wafers and I was very interested in your paper. I'm interested and grateful for the additional suggestions I've presented.

Please check the following points again.

1) Line 363~364: Could you please check the flow velocity unit (m2/s) ? I think it will be m/s.

2) Line 370& 386: Ø 564 µm ---> use different expression instead of Ø

3) Line 397& 400:  Miller et al.--->Miller et al.[19],  Gelber et al. ---> Gelber et al.[21]